# Single super-vortex as a proxy for ocean surface flow fields

**Imre M. Jánosi**[1,2,3], **Miklós Vincze**[3,4], **Gábor Tóth**[1], and **Jason A. C. Gallas**[2,5,6]

[1]Department of Physics of Complex Systems, Eötvös Loránd University, Pázmány Péter s. 1/A, H-1117 Budapest, Hungary
[2]Max Planck Institute for the Physics of Complex Systems, Nöthnitzer Str. 38, 01187 Dresden, Germany
[3]von Kármán Laboratory for Environmental Flows, Eötvös Loránd University, Pázmány Péter s. 1/A, H-1117 Budapest, Hungary
[4]MTA-ELTE Theoretical Physics Research Group, Pázmány Péter s. 1/A, 1117 Budapest, Hungary
[5]Complexity Sciences Center, 9225 Collins Avenue Suite 1208, Surfside, FL 33154, USA
[6]Instituto de Altos Estudos da Paraíba, Rua Silvino Lopes 419-2502, 58039-190 João Pessoa, Brazil

**Correspondence:** Imre M. Jánosi (imre.janosi@ttk.elte.hu)

**Abstract.** Empirical flow field data evaluation in a well studied ocean region along the U.S. West Coast revealed a surprisingly strong relationship between the surface integrals of kinetic energy and enstrophy (squared vorticity). This relationship defines a single isolated Gaussian super-vortex, whose fitted size parameter is related to the mean eddy size, and the square of the fitted height parameter is proportional to the sum of the square of all individual eddy amplitudes obtained by standard vortex census. This finding allows a very effective coarse-grained eddy statistics with minimal computational efforts. As an illustrative example, the westward drift velocity of eddies is determined from a simple cross correlation analysis of kinetic energy integrals.

## 1 Introduction

Mesoscale eddies (MEs) are energetic, swirling, time-dependent circulatory flows on a characteristic scale of around 100 km (see Fig. 1), which are observed almost everywhere in satellite altimetry data of global sea surface height (Chelton et al., 2007, 2011). The total volume transport by drifting eddies is comparable in magnitude to that of the large-scale wind-driven and thermohaline circulations (Zhang et al., 2014), therefore MEs play a crucial role in global material and heat transport and mixing of oceans. In spite of their importance, it is far from trivial to identify and characterize MEs from remote sensing data.

The vast majority of the ME studies is based on some automatic algorithm that identifies and tracks the eddies from gridded maps of sea level anomaly (SLA). Various Eulerian methods were developed and deployed in practice such as detecting closed contours of SLA (Chelton et al., 2011; Mason et al., 2014; Li et al., 2016; Schütte et al., 2016; Pessini et al., 2018), evaluating the geometry of the velocity vectors (Nencioli et al., 2010), determining contours of the Okubo-Weiss parameter (Chelton et al., 2007; Kurian et al., 2011; Ubelmann and Fu, 2011; Schütte et al., 2016; Pessini et al., 2018), or using wavelet analysis to identify coherent eddy-like structures (Rubio et al., 2009; Pnyushkov et al., 2018). Critical comparisons show that none of the Eulerian methods is superior to another (Souza et al., 2011; Escudier et al., 2016). The algorithms based on searching for finite-time Lagrangian coherent structures obey a better theoretical foundation (Haller, 2015; Beron-Vera et al., 2018; Haller et al., 2018), nevertheless a recent test of twelve different approaches revealed that the various methods often produce very different predictions for coherent structures. In addition, false positives and negatives can be produced too (Hadjighasem et al., 2017). Apart from the difficulties of identifying MEs, Amores et al. (2018) pointed out that the spatial resolution of gridded fields is also a critical limiting factor. It is not surprising that small vortices are detected in large numbers at fine grid sizes. However, it is somewhat unexpected that many large eddies remain unidentified by close contour searching when the velocity field is represented at lower resolutions (Amores et al., 2018).

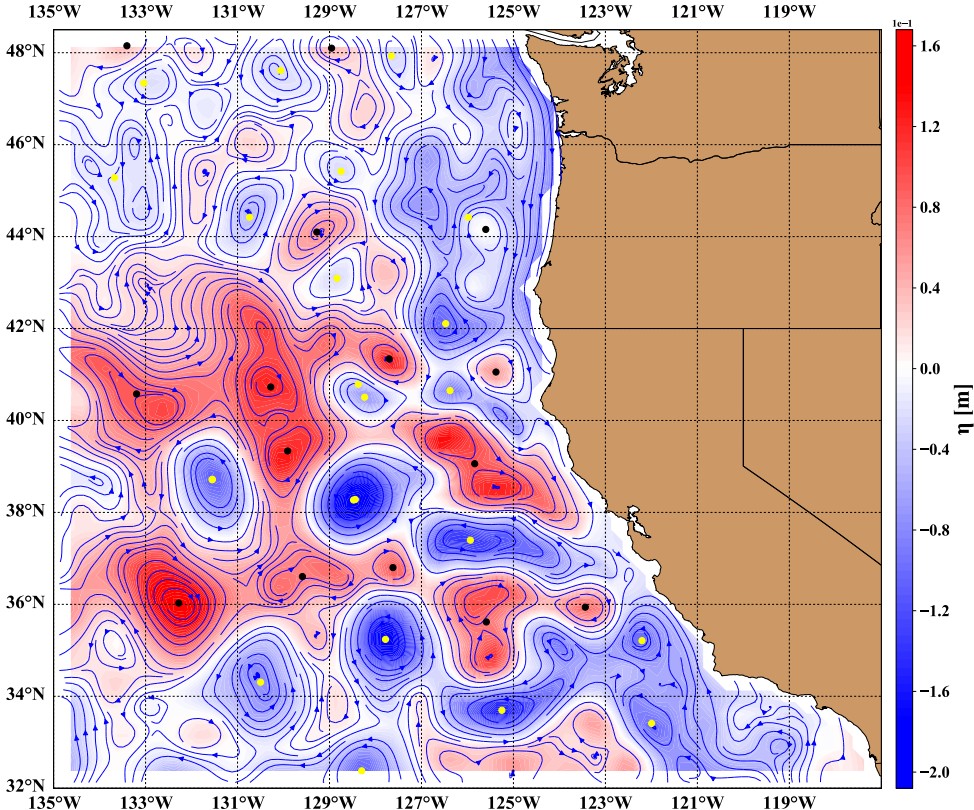

**Figure 1.** Visualization of the geostrophic flow field on a randomly choosen day (13 Oct 2013) from the data set over the U.S. West Coast by Risien and Strub (2016). Sea level anomalies ($\eta$) are color coded, blue stream lines indicate flow directions. The centers of cyclonic (yellow dots) and anticyclonic (black dots) eddies are determined by a standard algorithm (Chelton et al., 2011).

The original aim of our work was a detailed analysis of kinetic energy budget of the oceanic surface flow field along the U.S. West Coast. At the evaluation of integrated kinetic energy and enstrophy (squared vorticity), we found a non-trivial strong temporal correlation between these quantities. Since the dominating flow features are obviously mesoscale eddies (Fig. 1), it is rather straightforward to formulate an explanation related to the description of individual ocean vortices. One of the basic models is the Gaussian geostrophic vortex exhibiting the attractive features of finite total energy and total enstrophy over an infinite domain, and a simple closed relationship between them. We demonstrate here that a single Gaussian super-vortex properly describes the empirical energy/enstrophy ratio over an extended region, furthermore the height and radius of such super-vortex are strongly related to the mean values over the same area obtained by classical vortex census.

## 2 Shielded Gaussian vortices

As for the shape of ocean MEs, the common picture is that they are close to Gaussian humps or troughs (Hopfinger and van Heijst, 1993; Chelton et al., 2011). A detailed fitting pro-

cedure of about five million SLA profiles by Wang et al. (2015) revealed that around 50% of MEs are indeed Gaussian, another ∼40% are Gaussian over a sloping background or merger of two close Gaussian eddies, and the rest have a quadratic core resembling Rankine vortices. An isolated Gaussian circular eddy in geostrophic equilibrium (where the hydrostatic pressure gradient force is balanced by the local Coriolis force) can be characterized by the following radial profiles of height $\eta$, tangential velocity $v$, and vertical vorticity $\xi$ (in cylindrical coordinates):

$$\eta(r) = \eta_0 \exp\left(-\frac{r^2}{2R^2}\right) \ , \tag{1}$$

$$v(r) = -\frac{\eta_0 g}{fR^2} r \exp\left(-\frac{r^2}{2R^2}\right) \ , \tag{2}$$

$$\xi(r) = \frac{\eta_0 g}{fR^2}\left(\frac{r^2}{R^2} - 2\right) \exp\left(-\frac{r^2}{2R^2}\right) \ . \tag{3}$$

Here, $\eta_0$ and $R$ are the height and size parameters for the vortex, respectively, $g$ is the gravitational acceleration, and $f = 2\Omega \sin(\varphi)$ is the local Coriolis parameter at latitude $\varphi$ with $\Omega = 7.292 \times 10^{-5}$ s$^{-1}$ for the Earth. The label "shielded" in the title of this section refers to that the core of such a vor-

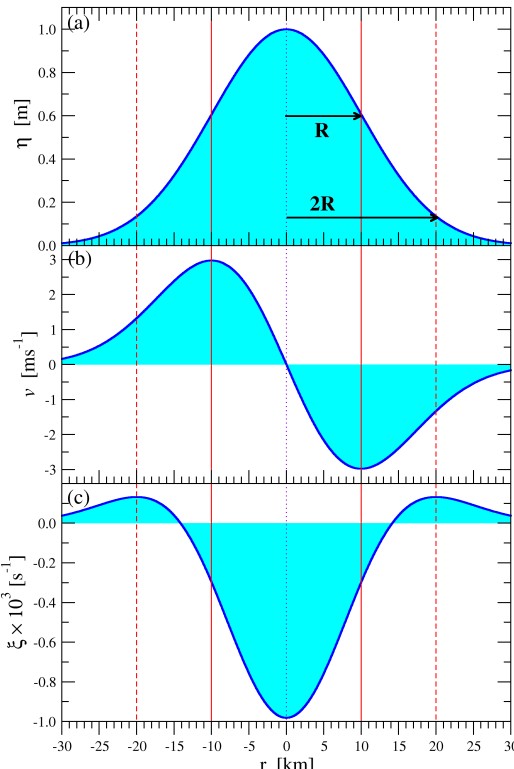

**Figure 2.** Characteristics of a shielded Gaussian geostrophic vortex with peak height $\eta_0 = 1$ m and size parameter $R = 10$ km at an approximate location of $45°$N latitude (Coriolis parameter $f = 10^{-4}$ s$^{-1}$). **(a)** Amplitude, see Eq. (1), **(b)** tangential velocity, see Eq. (2), and **(c)** vertical vorticity, see Eq. (3), as a function of radial distance $r$. Note that $R$ is the radial distance of maximum tangential velocity (vertical red line), and $2R$ is the distance of maximal vorticity in the shielding ring (vertical red dashed line). The "visual" radius based on closed contours of zero height anomaly is around 2.5 - 3 $R$.

tex is surrounded by a ring of opposite vorticity (Tóth and Jánosi, 2015), see Fig. 2c.

The simplest model of planetary-scale dynamics of the ocean is a single layer of homogeneous fluid, described by the two-dimensional (2D) barotropic Navier-Stokes equations in a co-rotating frame of reference (Bracco et al., 2004). In the absence of dissipative processes, such a model conserves the total kinetic energy $\iint KE = \frac{1}{2}\iint v^2 dA$, and total enstrophy $\iint Z = \frac{1}{2}\iint \xi^2 dA$. An appealing property of an isolated Gaussian vortex is that its total kinetic energy and enstrophy are finite over an infinite domain of integration:

$$I_{KE} = \frac{1}{2}\int_0^\infty 2\pi r v^2(r)dr = \frac{g^2\pi\eta_0^2}{2f^2} \ , \qquad (4)$$

$$I_Z = \frac{1}{2}\int_0^\infty 2\pi r \xi^2(r)dr = \frac{g^2\pi\eta_0^2}{f^2R^2} \ . \qquad (5)$$

Note that the total kinetic energy integral $I_{KE}$ depends only on the height parameter $\eta_0$, reflecting self-similarity in the velocity field, and that the ratio of the two integrals is simply $I_{KE}/I_Z = \frac{1}{2}R^2$. The very relationship was utilized in a recent paper by Li et al. (2018), in a different context of studying viscous decay of individual MEs.

## 3 Data analysis

Simple visual inspection of a reconstructed geostrophic flow field (Fig. 1) reveals that MEs are indeed the dominating features. The area shown in Fig. 1 is an extremely well-studied region of the California Current System (CCS) both by observations and calibrated high resolution numerical simulations (Kelly et al., 1998; Strub and James, 2000; Marchesiello et al., 2003; Castelao et al., 2006; Stegmann and Schwing, 2007; Capet et al., 2008a, b; Checkley and Barth, 2009; Matthews and Emery, 2009; Kurian et al., 2011; Molemaker et al., 2015; Yuan and Castelao, 2017). Openly available data compiled by Risien and Strub (2016) comprise a set of fields of sea level anomalies by combining gridded daily altimeter fields with coastal tide gauge data (Saraceno et al., 2008). The geographic area covers $32.0°$N – $48.5°$N (latitude) and $135.0°$W – $111.25°$W (longitude) with a spatial resolution of $0.25° \times 0.25°$. Daily mean geostrophic velocity fields are produced for the period 1 January 1993 - 31 December 2014 (8035 days). The primary validation compares geostrophic velocities calculated from the SLA values and velocities measured at four mooring sites in the test region (Risien and Strub, 2016).

Figure 3 illustrates the total enstrophy (squared vorticity) and total kinetic energy (sum of squared velocity components) integrated over the offshore region (see the dashed frame in Fig. 4) for each day of the record. The correlation is strikingly strong, and it is not trivial. When the shore region is included, much larger differences appear, especially when the area of integration is restricted to a narrow band along the shoreline. Fig. 5a clearly demonstrates that large correlation coefficients require large enough areas of integration, a value of 0.95 is reached around $A = 2.7 \times 10^5$ km$^2$ ($\sim 20^2$ grid cells or $5° \times 5°$). Nevertheless the geometry of the area must not be a square. The red and black symbols in Fig. 5a belong to meridional stripes of width of $1°$ and $2°$ longitudes (smaller areas are stripes eastward from $125.0°$W where the meridional length is restricted by the land). Their apparent scatter, however, is not random, the correlation coefficients in equal areas of integration (symbols lined up vertically in Fig. 5a) systematically increase with the distance from the shoreline.

By exploiting the strong correlations, the ratio of integrated kinetic energy and integrated enstrophy provides an effective size parameter of a hypothetical Gaussian supervortex as $R^{\text{eff}} = \sqrt{2\iint KE/\iint Z}$. Results for the temporal mean values of this quantity are shown in Fig. 5b. Note that

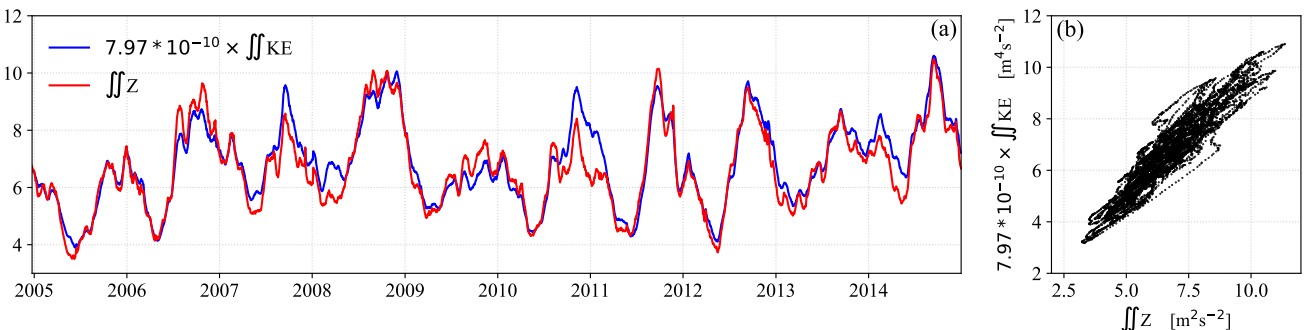

**Figure 3. (a)** Ten years of daily values for total enstrophy (red) and rescaled total kinetic energy (blue) integrated over the offshore region (westward from 125.0°W longitude, see Fig. 4), and **(b)** correlation plot of the two quantities. The rescaling factor for the kinetic energy integral is $7.97 \times 10^{-10}$ (see text).

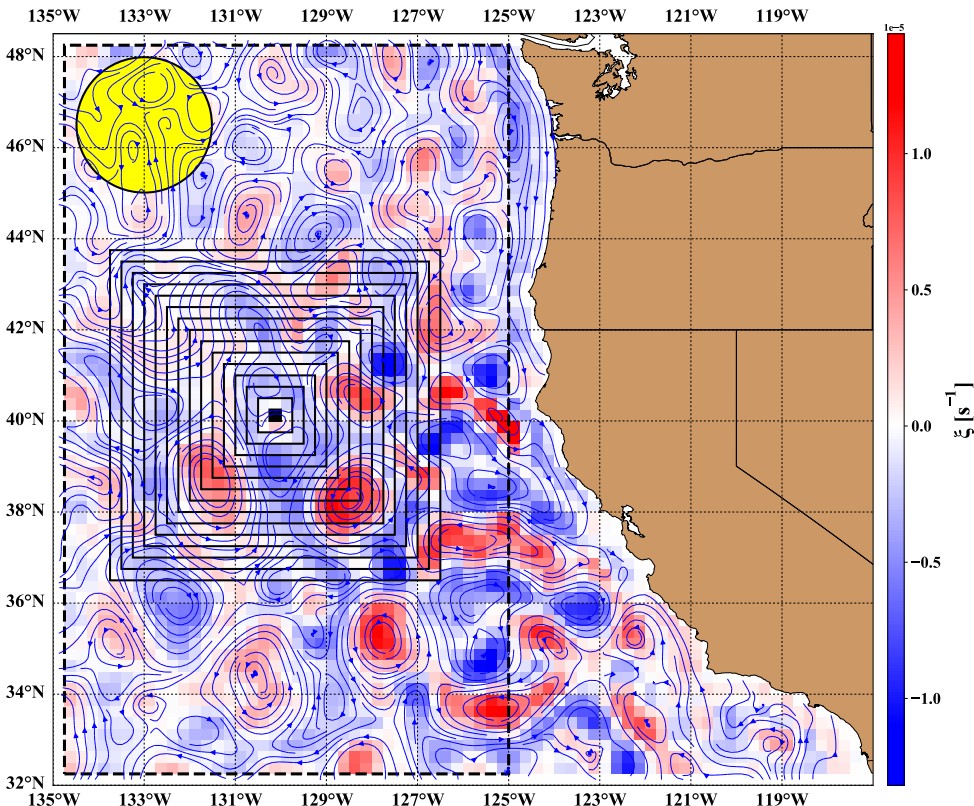

**Figure 4.** Visualization of the geostrophic flow field on the same day as in Fig. 1 (13 Oct 2013) from the data set over the U.S. West Coast by Risien and Strub (2016). Empirical vertical vorticity ($\xi$) is color coded, blue stream lines indicate flow directions. The color-mesh illustrates well the spatial resolution. Heavy dashed frame indicates the offshore region, where the integrated quantities in Fig. 3 are determined, and the yellow circle demonstrates the size of the hypothetical "super-vortex" related to mean vortex statistics on the given day over the offshore region (see text). Black squares illustrate the first 15 growing integration frames centered at the location 40.125°N, 130.125°W (see Fig. 5).

the obtained $R^{\mathrm{eff}} \approx 50$ km scale belongs to the $1\sigma$ width of a Gaussian profile given by Eq. (1). A visual contour of the super-vortex on a SLA map would have a radius closer to $\sim$ 2.5-3$R^{\mathrm{eff}} \approx 125$-150 km (see Figs. 2a and 4).

5    As for the height parameter of the super-vortex, Eq. (4) is used for an estimate of $\eta_0^{\mathrm{eff}}$ shown in Fig. 5c. Since it is

obtained from the total kinetic energy integrated over various areas $A$, an appropriate comparison requires a proper normalization. A practical choice correcting somewhat shape differences is the characteristic length scale $L = \sqrt{A}$. The error bars are much larger than the ones in Fig. 5b as a consequence of the marked annual oscillations shown in Fig. 3a.   10

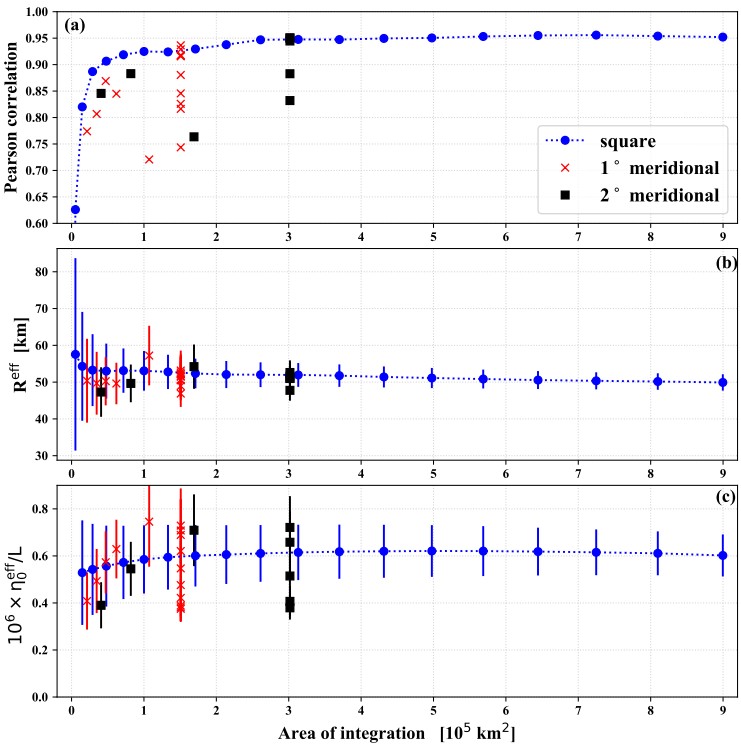

**Figure 5. (a)** Pearson correlation coefficient for the total kinetic energy and enstrophy as a function of the area of integration. Blue circles indicate growing correlations for square shaped areas around a central grid cell in the offshore region (40.125°N, 130.125°W), see Fig. 4. Red crosses (black squares) denote correlation coefficients for meridional stripes of width of 1° (2°) longitude. **(b)** Fitted mean scale parameter $R^{\mathrm{eff}}$ for a super-vortex determined from the ratio of integrated kinetic energy and enstrophy (in units of km). Notations are the same as in (a). **(c)** Fitted mean height parameter $\eta_0^{\mathrm{eff}}$ normalized by the square root of the area of integration $L$ (and rescaled for the sake of convenience) for a super-vortex determined from the integrated kinetic energy, see Eq. (4). Notations are the same as in (a).

These oscillations are canceled when the ratio of strongly correlated kinetic energy and enstrophy is considered. Similarly to the correlation coefficients in Fig. 5a, the fitted height values of $\eta_0^{\mathrm{eff}}$ for the meridional stripes (red crosses and black squares) exhibit systematic changes with the distance from the shoreline, as discussed below.

## 4 Eddy census

The super-vortex fit makes only sense when the parameters have some relationship with the existing MEs. In order to make such a comparison, we implemented the eddy census procedure of Chelton et al. (2011) based on closed SLA contour searches. The methodology is described in (Chelton et al., 2011; Oliver et al., 2015), here we emphasize three particular details. (i) The SLA fields in the data bank (Risien and Strub, 2016) exhibit marked annual oscillations, daily spatial mean values are changing between -8.6 and 10.1 cm. Since this range is comparable to the most common amplitude of the individual eddies (see below), we removed daily means before the eddy census. (ii) In order to avoid differences due to various definitions of the eddy amplitude, we adopted the

following rule: when the algorithm identified the location (lat, lon) of an eddy center, the amplitude value is imported directly from the (corrected) SLA field. (iii) We adopted the "equivalent radius" as scale parameter for an eddy (Chelton et al., 2011), that is $S = \sqrt{\pi^{-1} A_{tot}}$, where $A_{tot}$ is the total sum of grid cell areas identified inside a closed SLA contours.

Figure 6 shows the results of eddy census. The histograms are very similar to previous statistics at the same spatial resolution (Stegmann and Schwing, 2007; Chelton et al., 2011; Kurian et al., 2011; Amores et al., 2018). Note that the eddy scale histograms in Fig. 6a are sensitive to the level spacing parameter $\Delta l$ of the closed contour search, fine scale scans identify smaller eddies in a larger number. The oscillations at smaller eddy scales are due to the discretization error, the area of an eddy is composed of an integer number of grid cells. It is clear that the fitted super-vortex parameter $R^{\mathrm{eff}}$ fluctuates around the mean values of eddy scale histograms (black curve in Fig. 6a). We reiterate here that $R^{\mathrm{eff}}$ is an $1\sigma$ radius of a Gaussian vortex, while $S$ is closer to a "real" visual radius based on a closed contour estimate of zero height anomaly. As for the super-vortex height $\eta_0^{\mathrm{eff}}$, Fig. 6b illus-

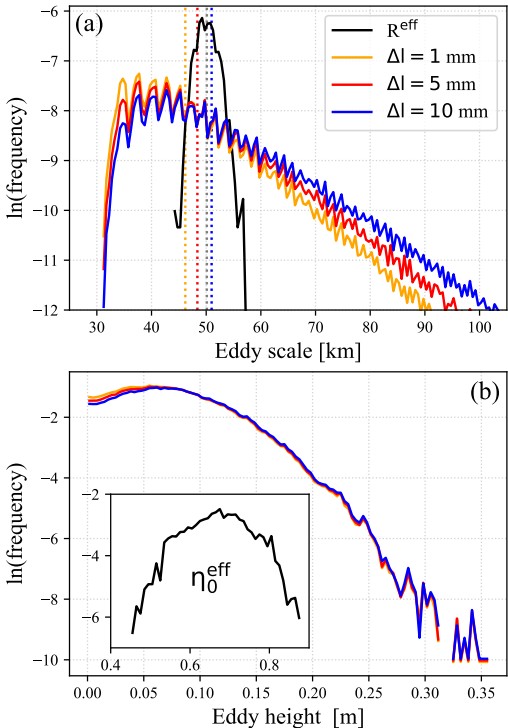

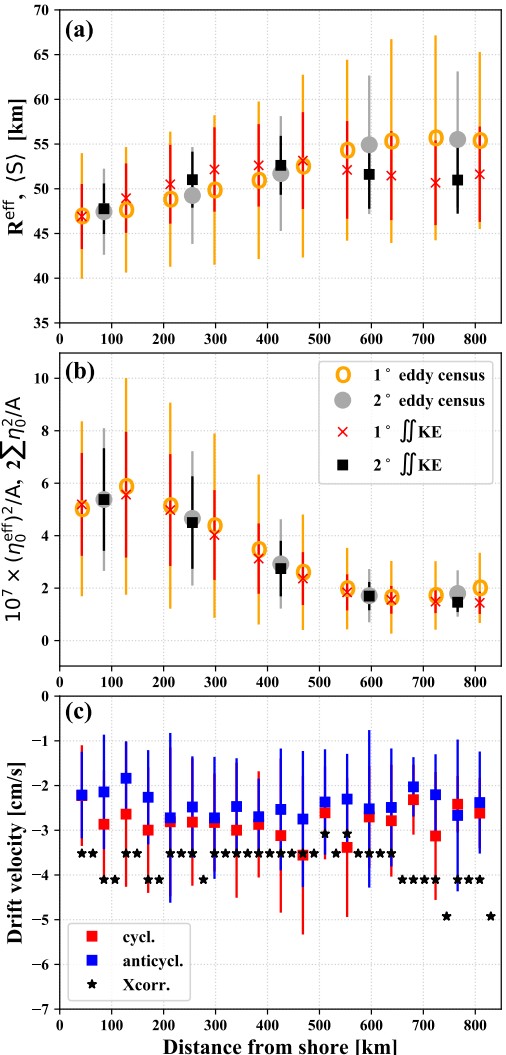

**Figure 6.** **(a)** Normalized eddy scale distributions obtained by individual eddy census with the closed contour SLA method (Chelton et al., 2011) at three different level spacing $\Delta l$, see legends. Vertical dotted lines indicate the mean values of the histograms. Black curve denotes the normalized histogram of $R^{\mathrm{eff}}$ parameter of the super-vortex. The logarithm of frequencies is scaled on the vertical scale. **(b)** Normalized eddy height distributions obtained by individual vortex census as in (a). The inset shows the histogram for the height parameter of the super-vortex fit $\eta_0^{\mathrm{eff}}$ in units of m. Both the eddy census and super-vortex fit were performed over the offshore region (westward from 125.0°W longitude, see Fig. 4).

**Figure 7.** **(a)** Fitted mean super-vortex radius $R^{\mathrm{eff}}$ and mean eddy scale $\langle S \rangle$ from eddy census, determined in meridional bands, and plotted as a function of mean distance from the shore. **(b)** Square of fitted mean super-vortex height $\left(\eta_0^{\mathrm{eff}}\right)^2$ and the sum of square of all individual eddy heights $\sum \eta_0^2$ normalized by the area of integration or eddy census $A$. **(c)** Estimated westward drift velocities by evaluating the cross correlation function Eq. (6), and from vortex tracking of MEs living at least 60 days.

trates that it is much larger than the height of individual eddies, as expected, because it is related to the total kinetic energy over the test area (the offshore region, in the particular case). For this reason, we compare the square of eddy ampli-
5 tudes in what follows.

The significant advantage of using super-vortex picture emerges when the fits are performed over sub-regions of the test area. We have shown already results for meridional stripes of widths of 1° and 2° in Figs. 5b and 5c. Figure 7a
10 illustrates local mean $R^{\mathrm{eff}}$ values compared with local mean eddy scale $\langle S \rangle$ as a function of the mean distance from the shore. Both quantities exhibit very good agreement and a clear tendency of growth when eddies move away from the shore. The error bars reflect temporal fluctuations over the
15 whole period of 8035 days which are much larger for the eddy census data, because their frequency fluctuates strongly day to day in a given narrow meridional band.

Figure 7b is a comparison of the height parameters of the super-vortex and eddy census. It is reasonable to consider a relationship between $\left(\eta_0^{\text{eff}}\right)^2$ and $\sum \eta_0^2$. The former measures the total kinetic energy [see Eq. (4)], while the latter is proportional to the sum of kinetic energies of all individual eddies when we assume that all of them are Gaussian vortices. The sum of kinetic energies based on $\sum \eta_0^2$ agrees pretty well with direct counting, when the kinetic energy is determined by adding up squared velocity components for each grid cell assigned to an eddy. Fig. 7b illustrates that an empirical ratio of around 2 arises in each meridional stripe, that is the long term mean value of kinetic energy for individually identified eddies is $\sim 50\%$ of the total kinetic energy in the test region. Interestingly, Amores et al. (2018) reported on a partition ratio between 1 and 5 fluctuating strongly in time, however they note that the total kinetic energy obtained for satellite altimetry accounts only for half of the real value. The tendency of initial growth upto $\sim 150$ km (see Fig. 7b) might be related to the fact that eddies are generated mostly along the shore, and later they are slowly decaying during the drift in open water.

A well-known characteristic of eddy trajectories is the strong tendency for purely westward propagation (Cushman-Roisin et al., 1990; Chelton et al., 2007, 2011; Kurian et al., 2011). Chelton et al. (2007) found globally that only about 0.25% of the eddies have mean drift directions that deviated by more than $10°$ from pure zonal, however Kurian et al. (2011), and Stegmann and Schwing (2007) obtained stronger dispersion in the CCS study area. Together with the traditional eddy tracking algorithm, we used our approach to evaluate the cross correlations of total kinetic energy $I(t) = \frac{1}{2} \iint v^2 dA(t)$ between neighboring meridional bands of width of a single grid cell (0.25°):

$$X(\tau) = \frac{\langle [I(t)_i - \bar{I}_i][I(t \pm \tau)_{i-1} - \bar{I}_{i-1}] \rangle}{\sigma_i \sigma_{i-1}} \ , \tag{6}$$

where the time lag $\tau$ represents a temporal shift between the two time series by $\tau$ days, overbar denotes temporal mean, and $\sigma$ is the standard deviation in the given band. Indeed, we find clear maxima at nonzero time lags (actual values are between 5 and 8 days) indicating that total kinetic energy and enstrophy are mostly advected in the offshore region, production or loss is almost negligible (considering geostrophic flow). The time lag and distance of neighboring bands permit an easy estimate of westward drift velocities, the results are shown in Fig. 7c. Drift velocity values in the literature are in the same order of magnitude (Stegmann and Schwing, 2007; Kurian et al., 2011; Chelton et al., 2007, 2011), similarly to our test. As for a direct validation, all individual eddy tracks are evaluated which had longer lifetime than 60 days (432 cyclonic and 422 anticyclonic MEs are identified). The cut at 60 days is somewhat arbitrary, however we think that the detection error from both the limited spatial and temporal resolutions is larger for short living vortices (note that the typical westward traveling distance during 60 days is $\sim 155$ - 200 km). Drift values estimated from vortex tracking belong to the centers of eddies, and as expected from a stable $\beta$-drift, no spatial dependence in the zonal direction is revealed. Theoretical considerations suggest that anticyclonic eddies might drift faster than cyclonic ones (Cushman-Roisin et al., 1990), however we could not detect statistically significant difference between the two subgroups of trajectories.

## 5 Conclusions

We proposed a simple description of geostrophic ocean surface flow fields by exploiting the following results. Firstly, a shielded Gaussian vortex has a finite total kinetic energy and finite total enstrophy, the ratio of them is proportional to the square of the radius of the vortex. Secondly, these two quantities determined from empirical velocity data are strongly correlated, and their ratio correlates with the mean eddy size obtained from traditional eddy census. Thirdly, the fitted amplitude parameter is strongly related to the sum of all squared eddy amplitudes. While this description cannot replace traditional eddy census algorithms, it is certainly able to extract coarse grained eddy statistics in order to follow temporal and regional changes of eddy activity.

*Author contributions.* I.M.J. designed research; I.M.J. and M.V. performed research; G.T. and J.A.C.G. contributed new numerical/analytical tools; I.M.J. and G.T. analyzed data; and I.M.J., M.V. and J.A.C.G. wrote the paper.

*Competing interests.* The authors declare no conflict of interest.

*Acknowledgements.* This work was supported by the Hungarian National Research, Development and Innovation Office under grant numbers FK-125024 and K-125171, and by the Max-Planck Institute for the Physics of Complex Systems in the framework of an Advanced Study Group on "Forecasting with Lyapunov Vectors". J.A.C.G. was supported by CNPq, Brazil.

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
