# Peer review of "Single super-vortex as a proxy for ocean surface flow fields"

_Ocean Science, 2019_

## Referee Comment (RC1) · Anonymous Referee #1 · 30 May 2019

In this work the authors focused on a well-studied area along the U. S. West Coast, where the mesoscale eddies are the dominating features. At first, they described the characteristics of an ideal Gaussian isolated vortex, highlighting the relationship between the total kinetic energy and the enstrophy in absence of dissipative forces. After that, the authors make use of daily gridded Sea Level Anomaly data for a period of 8035 days to show the time series of the total kinetic energy and the enstrophy. They found a not trivial strong correlation between the two quantities, which increases in function of the dimension of the area of integration and with the distance from the coast. From the relationship between the energy and the enstrophy, they extract the effective size of a hypothetical Gaussian super vortex which in some way may constitute a model for the eddies. The comparison between the hypothetical super vortex and the existing ed-

dies has been computed thanks to an eddy geometrical census procedure developed by Chelton et al. (2011). In conclusion, the authors proposed a simple description of the geostrophic ocean surface flow fields, highlighting the fact that a shielded Gaussian super vortex has finite total kinetic energy and enstrophy and the ratio of them is proportional to the square of the radius of the vortex. Furthermore, they calculated the two quantities from altimetry data and computed the mean eddy size, which was comparable with the radius computed from the traditional eddy census. The method implemented cannot substitute the traditional eddy detection algorithms, but can be useful to extract coarse grained statistics. Furthermore, the authors computed, as an illustrative example, the westward drift velocity of eddies from a simple cross correlation analysis of kinetic energy integrals.

The paper is well structured and the results are original. I think it can be very relevant for the international community.

I just suggest some minor remarks:

Pg 1 ln 15: You can add "some exceptions to the remote sensing of eddies are the in situ description of an anticyclone in the North Atlantic by Martin and Richards (2001) and the sampling of an anticyclone in the Algerian basin along its main axes by Cotroneo et al. (2016)"

Pg 2 ln 17: You should add the aim of this work that is missing in the section "Introduction"

Fig 1: add the geographical references, the square and the stripes of integration and finally the "visual contour" of the super vortex (see pg 4 ln 17).

Pg 3 ln 11: Why the core of such a vortex is surrounded by a ring of opposite vorticity? Add references or explain better

Pg 8 ln 23: algorithm

Pg 8 ln 33: why did you chose 60 days? Please, provide a reason.
In general, I suggest in the future (not for this work) to test your method in other regions, where the properties of the eddies and the altimetry data may be have different characteristics.

References:

Martin, A. P. & Richards, K. J. Mechanisms for vertical nutrient transport within a North Atlantic mesoscale eddy. Deep Sea Res. Part II Top. Stud. Oceanogr. 48, 757–773 (2001)

Cotroneo, Y. et al. Glider and satellite high resolution monitoring of a mesoscale eddy in the algerian basin: Effects on the mixed layer depth and biochemistry. J. Mar. Syst. 162, 73–88 (2016).

---

## Referee Comment (RC2) · Anonymous Referee #2 · 30 May 2019

The authors studied the eddies along the U.S. West Coast. They revealed a surprisingly strong relationship between the surface integrals of kinetic energy IKE and enstrophy Iz (squared vorticity) with the observation data. In addition, they also noted that the square of the fitted height parameter is proportional to the sum of the square of all individual eddy amplitudes obtained by standard vortex census. In general, the paper is well written. However, there are still some issues to be clarified.

Major comments: The Relationship of IKE vs Iz in case of Gaussian shape vortex is not surprise for me, which was previously noted in Li et al. (2018). They used area instead of $1/2R^2$, then applied to census of global ocean eddies without any validation [Li et al., 2018]. It is interesting that the direct integration of oceanic data (Figure 2) supports this relationship. Nevertheless, this notation should be clearly presented as

[Figure]

this study has done. Please add a short notation addressing this issue after equation (5) in section 2.

Figure 3. I don't understand what Red crosses (black squares) mean from the caption of figure. For example, why there are so many red crosses at a given area, e.g., 3.0*10ˆ5 km2, what's the difference? Even I have read the explanations in Lines 10-13, page 4. The authors should add both some notations in figure caption and some explanations in result. The result in Figure 3c implies that there is a linear relationship between the eddy amplitude and the eddy scale. The larger the eddy is, the higher the amplitude is. Authors may want to address this in revision.

Figure 4. I suggest authors trying eddy area other than eddy scale in Figure 4a, since authors have already noted that area is an important parameter in the study. The result of eta_0ˆeff (∼0.7 m) in average may be 10 times of that obtained by eddy identification method (e.g., Chelton et al., 2011; Li et al., 2016), which can be also seen from histogram for eddy height in Figure 4b, where the height peaks at about 0.07 m. So I simply suspect that authors might incidentally make some mistake for this parameter by ignoring the gravitational acceleration g in the calculation.

Figure 5. It is surprising that the difference between eddy radius and eddy scale minimizes at near shore region, but maximizes at off shore region. Could authors address more about this?

Figure 5. The square of the fitted height parameter is proportional to the sum of the square of all individual eddy amplitudes. Could authors go further to find a simple relation between them, like IKE/IZ = 1/2Rˆ2?

The cyclonic eddies have relatively faster westward propagation speeds than anticyclonic eddies, which is seldom mentioned as far as I know. Could authors make some further explanations?

Minor comments: Figure 1. Add the point 40N, 130W in this figure with a notable

symbol. Figure 2. Add the coefficient of two curves in the figure, if possible.

Typo: Page 8, line 16. Reports->reported

References: Chelton, D. B., Schlax, M. G., and Samelson, R. M. (2011): Global observations of nonlinear mesoscale eddies, Progr. Oceanogr., 91, 167-216, 2011. Li, Q.-Y., Sun, L. and Lin, S.-F. (2016) GEM: A Dynamic Tracking Model for Mesoscale Eddies in the Ocean. Ocean Science, 12, 1249-1267. Li, Q.Y., Sun, L. and Xu, C. (2018) The Lateral Eddy Viscosity Derived from the Decay of Oceanic Mesoscale Eddies. Open Journal of Marine Science, 8, 152-172.

---

## Author Comment (AC1) · 8 Jun 2019

We thank the Referee for the report and the supporting remarks. Please find here our responses to the minor comments. We attach the revised manuscript as Supplement, where two new figures and several changes (also detailed here) are inserted.

Referee remark (1): "Pg 1 ln 15: You can add "some exceptions to the remote sensing of eddies are the in situ description of an anticyclone in the North Atlantic by Martin and Richards (2001) and the sampling of an anticyclone in the Algerian basin along its main axes by Cotroneo et al. (2016)""

Response (1): After a pretty long consideration we decided not to insert the suggested papers. This is because they are loosely related to our work, which is primarily a

statistical analysis of an extended oceanic flow field. Many other studies focusing on individual vortices should be listed as "exeptions to the remote sensing", and we would like to limit the length of reference list to be tractable.

Referee remark (2): "Pg 2 ln 17: You should add the aim of this work that is missing in the section "Introduction""

Response (2): Following your suggestion, we inserted a whole paragraph to the end of Introduction as follows (Pg 2 ln 8): "The original aim of our work was a detailed analysis of kinetic energy budget of the oceanic surface flow field along the U.S. West Coast. At the evaluation of integrated kinetic energy and enstrophy squared vorticity), we found a non-trivial strong temporal correlation between these quantities. Since the dominating flow features are obviously mesoscale eddies (Fig. 1}), it is rather straightforward to formulate an explanation related to the description of individual ocean vortices. One of the basic models is the Gaussian geostrophic vortex exhibiting the attractive features of finite total energy and total enstrophy over an infinite domain, and a simple closed relationship between them. We demonstrate here that a single Gaussian super-vortex properly describes the empirical energy/enstrophy ratio over an extended region, furthermore the height and radius of such super-vortex are strongly related to the mean values over the same area obtained by classical vortex census."

Referee remark (3): "Fig 1: add the geographical references, the square and the stripes of integration and finally the "visual contour" of the super vortex (see pg 4 ln 17)."

Response (3): Following your suggestion, we inserted a new Figure (Fig. 4 in the revised manuscript), where we illustrate all the integration frames (apart from the vertical meridional stripes of width of 1 and 2 degrees), and the visual contour of the super-vortex. Probably it is better than pushing everything in Fig. 1.

Referee remark (4): "Pg 3 ln 11: Why the core of such a vortex is surrounded by a ring of opposite vorticity? Add references or explain better"

Response (4): This is simply a basic property of a Gaussian vortex. Nevertheless, in order to make this point clear, we inserted a new Figure (Fig. 2 in the revised manuscript), where we plot the height profile, tangential velocity and vertical vorticity Eqs. (1)-(3). Note that opposite vorticity does not mean a reversed circulation.

Referee remark (5): "Pg 8 ln 33: why did you chose 60 days? Please, provide a reason."

Response (5): As requested, we inserted an explanation for the cut of 60 days as follows (Pg 10 ln 27-29): "The cut of 60 days is somewhat arbitrary, however we think that the detection error from both the limited spatial and temporal resolutions is larger for short living vortices (note that the typical westward traveling distance during 60 days is $\sim$ 155 - 200 km)."

Finally, we thank for the general suggestion, we are currently working on the extension of the concept to different geographic locations.

Please also note the supplement to this comment:
https://www.ocean-sci-discuss.net/os-2019-14/os-2019-14-AC1-supplement.pdf

---

## Author Comment (AC2) · 9 Jun 2019

We thank the Referee for the report and the supporting remarks. Please find here our responses to the critical comments. We attached the revised manuscript as Supplement to the previous Response to Referee #1, we refer to its figure-page-line numbers here.

Referee remark (1): "Major comments: The Relationship of IKE vs Iz in case of Gaussian shape vortex is not surprise for me, which was previously noted in Li et al. (2018). They used area instead of $1/2R^2$, then applied to census of global ocean eddies without any validation [Li et al., 2018]. It is interesting that the direct integration of oceanic data (Figure 2) supports this relationship. Nevertheless, this notation should be clearly

presented as this study has done. Please add a short notation addressing this issue after equation (5) in section 2."

Response (1): Many thanks for pointing out the reference. We inserted the citation and mention it where it is requested (page 4, lines 19-20). Nevertheless we think that the context of Liu et al. (2018) is rather different, and not only the lack of validation is missing there.

Referee remark (2): "Figure 3. I don't understand what Red crosses (black squares) mean from the caption of figure. For example, why there are so many red crosses at a given area, e.g., 3.0*10ˆ5 km2, what's the difference? Even I have read the explanations in Lines 10-13, page 4. The authors should add both some notations in figure caption and some explanations in result."

Response (2): We inserted a new Figure (Fig. 4 in the revised manuscript) in order to better explain the integration frames. As for the red and black symbols in Fig. 3 (now Fig. 5), we clearly describe (page 5, lines 9-13) that they belong to equal-area meridional stripes of width of 1 and 2 degrees (vertical stripes on Figs. 1 and 4), the difference between them is their main distance from the shoreline.

Referee remark (3): "The result in Figure 3c implies that there is a linear relationship between the eddy amplitude and the eddy scale. The larger the eddy is, the higher the amplitude is. Authors may want to address this in revision."

Response (3): We do not entirely understand this remark. Fig. 3c (Fig. 5c in the revised version) refers to the "super-vortex" amplitude parameter obtained from Eq. (4). The linear behavior (or stable saturation after proper normalization shown in Fig. 5c) indicates only that the total kinetic energy is almost homogeneously distributed over the study area.

Referee remark (4): "Figure 4. I suggest authors trying eddy area other than eddy scale in Figure 4a, since authors have already noted that area is an important parameter in

the study. The result of eta_0ˆeff ($\sim$0.7 m) in average may be 10 times of that obtained by eddy identification method (e.g., Chelton et al., 2011; Li et al., 2016), which can be also seen from histogram for eddy height in Figure 4b, where the height peaks at about 0.07 m. So I simply suspect that authors might incidentally make some mistake for this parameter by ignoring the gravitational acceleration g in the calculation."

Response (4): We do not think that 'g' is missing from our estimate. As explained in details, eta_0ˆeff ($\sim$0.7 m) is derived from the total kinetic energy integrated over an extended region (direct sum of velocity component squares), see Eq. (4). It can easily be higher by a factor of 10 than the mean height of individual eddies. Actually, as it is pointed out, the sum of squared eddy heights is related to eta_0ˆeff.

Referee remark (5): "Figure 5. It is surprising that the difference between eddy radius and eddy scale minimizes at near shore region, but maximizes at off shore region. Could authors address more about this?"

Response (5): Thanks for this remark, it is an interesting point. We think that this behavior is related to the fact that the near shore region is the main place of mesoscale eddy formation, which is a complicated process including wind forcing, interaction with the California Current System, rough shoreline effects, bottom friction in the shallow regions, etc. At the moment we have not enough knowledge on the transient phases of ME formation, nevertheless the height of an eddy must have an increasing phase at the beginning. We need more detailed analysis to go beyond speculations.

Referee remark (6): "Figure 5. The square of the fitted height parameter is proportional to the sum of the square of all individual eddy amplitudes. Could authors go further to find a simple relation between them, like IKE/IZ = 1/2Rˆ2?"

Response (6): We do not entirely understand this remark. The first quantity is derived from the total kinetic energy integrated over an extended region (direct sum of velocity component squares). The second quantity is derived from vortex census, amplitudes are from SLA values. Their stable ratio is around 2 (Fig. 7b), and we do not know

where to go further.

Referee remark (7): "The cyclonic eddies have relatively faster westward propagation speeds than anticyclonic eddies, which is seldom mentioned as far as I know. Could authors make some further explanations?"

Response (7): As requested, we inserted a related statement (p. 10, l. 30-32 in the revised version) as follows: "Theoretical considerations suggest that anticyclonic eddies might drift faster than cyclonic ones (Cushman-Roisin et al., 1990), however we could not detect statistically significant difference between the two subgroups of trajectories."

Referee remark (8): "Minor comments: Figure 1. Add the point 40N, 130W in this figure with a notable symbol."

Response (8): Following the suggestions of both Referees, we inserted a new Figure (Fig. 4 in the revised manuscript) to illustrate all the integration frames and locations.

Referee remark (9): "Figure 2. Add the coefficient of two curves in the figure, if possible."

Response (9): We are sorry, we cannot understand this request. The rescaling factor of integrated kinetic energy is clearly printed in both panels, and repeated in the caption.

The typo is corrected, thanks.

---

## Author Response (AR1)

**Author's Response**

Since we did not receive any further comment on our manuscript, please find our detailed responses above directly at the Referees' Reports. All changes in the revised version are denoted by blue typesetting.

[revised manuscript text omitted]